# Mouthwash-Based Highly Sensitive Pyro-Genotyping for Nine Sexually Transmitted Human Papilloma Virus Genotypes

**DOI:** 10.3390/ijms21103697

**Published:** 2020-05-24

**Authors:** Yoshiyuki Watanabe, Yukiko Seto, Ritsuko Oikawa, Takara Nakazawa, Hanae Furuya, Hidehito Matsui, Sachiko Hosono, Mika Noike, Akiko Inoue, Hiroyuki Yamamoto, Fumio Itoh, Kota Wada

**Affiliations:** 1Department of Internal Medicine, Kawasaki Rinko General Hospital, Kawasaki 210-0806, Japan; setoyukiko@mac.com (Y.S.); takara0805@icloud.com (T.N.); hanae.furuya@med.toho-u.ac.jp (H.F.); hidehito.matsui@med.toho-u.ac.jp (H.M.); s-hosono@med.toho-u.ac.jp (S.H.); mika.doi@med.toho-u.ac.jp (M.N.); akiko.inoue@med.toho-u.ac.jp (A.I.); 2Department of Otolaryngology, Toho University Omori Medical Center, Tokyo 143-8540, Japan; r-oikawa@marianna-u.ac.jp (R.O.); h-yama@marianna-u.ac.jp (H.Y.); fitoh@marianna-u.ac.jp (F.I.); 3Division of Gastroenterology and Hepatology, Department of Internal Medicine, St. Marianna University School of Medicine, Kawasaki 216-8511, Japan; wadakota@med.toho-u.ac.jp

**Keywords:** HPV genotyping, mouthwash, sexually transmitted infections (STIs), pyrosequencing

## Abstract

Human papillomavirus (HPV) is a common sexually transmitted infection worldwide, which spreads via contact with infected genital, anal, and oral/pharyngeal areas (oral sex) owing to diverse manners of sexual intercourse. In this study, we devised an oral HPV detection method using mouthwash waste fluids that causes less psychological resistance to visiting the outpatient otolaryngology departments. We successfully detected only the specific unique reverse sequencing probe (using pyro-genotyping) and identified the nine genotypes of HPV targeted for vaccination by pyrosequencing the mouthwash waste fluids of non-head and neck cancer patient volunteers (*n* = 52). A relatively large number (11/52) of mouthwash waste fluids tested positive for HPV (21.2%; genotype 6, *n* = 1; 11, *n* = 1; 16, *n* = 1; and 18, *n* = 8). These results surpassed the sensitivity observed testing the same specimens using the conventional method (1/52, 1.9%). Our method (pyro-genotyping) was developed using nine HPV genotypes targeted for vaccination and the results were highly sensitive compared to those of the conventional method. This less expensive, high-throughput, and simple method can be used for detecting oral HPV infection with fewer socio-psychological barriers.

## 1. Introduction

More than one million people are infected with sexually transmitted diseases every day worldwide. Human papillomavirus (HPV), one of the causes of sexually transmitted infections (STIs), differs from the human immunodeficiency virus and herpes simplex virus genetically and clinically. Globally, more than 290 million women are infected with HPV [1,2].

HPV is clinically divided into 14 cancer high-risk genotypes (16, 18, 31, 33, 35, 39, 45, 51, 52, 56, 58, 59, 66, 68) whose epidemiological association with cancer has been indicated, and 5 cancer low-risk genotypes (6, 11, 42, 43, 44), which lead to benign lesions such as warts, genital condylomata, and laryngeal papillomata [3,4,5].

HPV infection is usually asymptomatic and transient. However, persistent HPV infection has been shown to progress from mild, moderate, and high-grade dysplasia to carcinoma in situ. High-risk mucosal HPV strains infecting the epithelia of the genitals, oral cavity, and laryngopharynx are detected not only in individuals with cervical cancer, but also in those with anal [6], penile [7], skin [8], and head and neck cancers [9].

Vaccination prevents HPV infection and the development of precancerous lesions. Approximately 60–70% cases of cervical cancer can be prevented with the approved 2-valent (HPV 16 and 18) or 4-valent (additionally, HPV 6 and 11) HPV vaccines. Recently, a 9-valent (containing additional antigens for HPV 31, 33, 45, 52, and 58) HPV vaccine that can prevent over 90% of the incidences of cervical cancer has been reported [10,11,12]. As part of numerous national vaccination programs, HPV vaccines are being administered to women in 71 nations (37%), while 11 (6%) also provide them to men [13,14,15]. In addition, the 4-valent and 9-valent HPV vaccines are also effective against the low-risk strains HPV 6 and 11, which is crucial for preventing STIs and the development of benign tumor lesions such as warts, genital condylomata, and laryngeal papillomata [16].

Unfortunately, HPV-infected individuals often refrain from check-ups because of psycho-social barriers, highlighting the necessity of developing unobtrusive and simple detection methods. Here, as a first step of the challenge, we aimed to devise an alternative diagnostic method for oral HPV genotyping using mouthwash waste fluids. This method is simple and less invasive, can be performed during otolaryngology visits, and is associated with low psychological resistance. This method would not only detect STIs contracted via contact with infected oral/pharyngeal areas, but also provide an opportunity for stress-free screening of women/men between the ages of 20 and 30 years and reduce the incidence of oral HPV-related health problems [17]. Our mouthwash waste fluid-based pyrosequencing method is less expensive, high-throughput, and simple, and it may be useful for detecting and genotyping HPV from the oral/pharyngeal areas.

## 2. Results

### 2.1. Volume and Quality of Genomic DNA Recovered from Mouthwash

To isolate genomic DNA from the mouthwash, 52 mixed samples (50 mL each) were collected from the waste tank. The total amount of genomic DNA in each sample was 61.24 ± 13.22 μg of DNA, with A260/280 ratio of 1.81 ± 0.88, analyzed using a NanoDrop ND-1000 spectrophotometer (Nano Drop Technologies, Wilmington, Del., USA).

### 2.2. Amplification and Direct Sequencing of the 18 HPV Genotype-Harboring Plasmids from HeLa Cells

We successfully confirmed the genotypes via direct sequencing of the DNA extracted from the 18 artificially constructed HPV genotype-bearing plasmid (6, 11, 16, 18, 31, 33, 35, 39, 40, 42, 43, 44, 45, 52, 56, 58, 59, 66) sequences, focusing on the sequences known to be downstream of the HPV L1 MY09/11 consensus primer site, and HeLa cells (genotype 18) (Figure 1A).

### 2.3. Amplification of Genes from HPV in the Clinical Mouthwash Waste Samples

Multiple HPVs were analyzed using the clinical mouthwash waste samples (Table 1). The DNA was successfully extracted from all HPV-positive samples and amplified using the primer sets for both direct sequencing and pyrosequencing (Figure 1A,C and Figure 2B).

### 2.4. Pyrosequencing Was Suitable for Genotyping Nine HPV Plasmids from HeLa Cells and Clinical Mouthwash Samples

Using the pyrosequencing approach (Figure 1C), we validated the high reproducibility of the genotype identification in nine plasmids carrying the HPV L1 region in HeLa cells (Figure 3). We also successfully genotyped HPV in all positive samples using the clinical mouthwash waste fluids via pyrosequencing (Figure 3).

### 2.5. Pyrosequencing Approach Was More Sensitive than the Conventional HPV-DNA Method

We successfully identified all 11 positive samples (11/52, 21.2%), which were amplified using the HPV L1 MY09/11 consensus primer set. Using genotyping, we identified their genotypes as follows: 6 (*n* = 1), 11 (*n* = 1), 16 (*n* = 1), and 18 (*n* = 8) (Figure 3, Table 1). In contrast, we were able to analyze only one sample (1/52, 1.9%) using the hybrid capture method (Figure 3, Table 1). Polymerase chain reaction (PCR) with restriction enzyme-based HPV genotyping detected only two samples (2/52, 3.8%, genotype 6; *n* = 0, 11; *n* = 0, 16; *n* = 1, and 18; *n* = 1). Using our pyrosequencing method, we analyzed the 52 clinical mouthwash samples with more sensitivity than the HPV-DNA method. Furthermore, all the nine HPV genotypes present in the vaccines were analyzed and genotyped using pyrosequencing (6, 11, 16, 18, 31, 33, 45, 52, and 58).

## 3. Discussion

This study focused mainly on assessing the technical methodology of our new “pyro-genotyping” using mouthwash samples of healthy volunteers. Mouthwash-based DNA sequencing is widely utilized by researchers. However, they use multi-primer sets for detecting each genotype of the virus. We can report being the first to use a unique reverse probe for the identification of nine genotypes of HPV. We discovered two key factors: one is the reverse reading using a unique reverse probe and the other is using the SEQ mode of pyrosequencing.

HPV infects epithelial cells of the skin and mucosa, causing local hyperplasia. Generally, it enters the body via minor wounds in the skin and mucosa caused by sexual activity [18,19,20], and it can be passed via sexual contact even when an infected individual is asymptomatic. However, vaccines can halt these health problems [2]. Nevertheless, the global rates of vaccination are not high, especially in Japan. In addition to the genital and anal areas, the mucosal membrane of the oropharynx can be infected with HPV, owing to the diversification in the methods of sexual intercourse in recent years [21,22]. In women, HPV is typically detected during gynecological examination; cervical HPV infections usually clear up without any intervention within a few months after acquisition, and approximately 90% cases are cured within 2 years.

In this study, we devised a method for the detection of oral HPV infection using mouthwash waste fluids that can be used in otolaryngology outpatient departments. In addition, we performed genotyping analysis based on the L1 gene region, which codes for the virus capsid protein of the HPV, using high-throughput pyrosequencing, which is less time-intensive and inexpensive than conventional direct sequencing. However, designing a unique probe for identifying all nine target genotypes using sequencing-based HPV genotyping analysis was challenging. We circumvented this problem using only the “reverse” sequencing probe (pyro-genotyping).

Verification tests using vectors for each HPV genotype showed that all nine HPV strains could be analyzed. Furthermore, a relatively high rate of HPV-positive cases, 11/52 (21.2%), where *n* = 52 was the number of bottles with mixed waste fluids from multiple patients, was observed in the mouthwash waste fluids (a mixture of waste fluids from different patients collected in the waste fluid bottles in the examination units of the outpatient department of otolaryngology). The genotype breakdown was as follows: genotype 6 (*n* = 1), 11 (*n* = 1), 16 (*n* = 1), and 18 (*n* = 8). This method was more sensitive than the conventional HPV-DNA method (hybrid capture method; 1/52, 1.9%) and the conventional genotyping method (PCR with restriction enzyme-based genotyping method; 2/52, 3.8%) [23,24]. Rosenthal et al. reported genotyping 14 HPV in oral rinse using real-time PCR-based HPV testing, but this assay did not include the 6 and 11 genotypes. Furthermore, their HPV test requires 12 different fluorescent DNA probes and 16 different primers, and identifies genotype 16, 18, or any others [25]. Fakhry et al. also genotyped HPV in oral rinse or swab samples using in situ hybridization or PCR, but they could only identify genotype 16 or any other oncogenic HPV. Our pyro-genotyping method detects not only the high-risk HPV genotype (specific), but also genotypes 6 and 11, using a unique primer and probe [26].

We could identify all nine HPV genotypes targeted by the HPV vaccines using the pyrosequencing-based genotyping method and a specific reverse unique probe [27,28,29]. The hybrid capture method used for comparison in the present study detects the moderate- to high-risk HPV strains, which cause cervical cancer. This method can identify the infection caused by 13 moderate- to high-risk HPV strains, but it cannot detect HPV 6 and 11.

In conclusion, we have developed a new oral HPV detection methodology (pyro-genotyping) using mouthwash waste fluid covering nine HPV genotypes (equal to the 9-valent HPV vaccine). It is a cost effective, high-throughput, and simple method for detecting oral HPV infection. However, our study does not provide sufficient evidence for the clinical use of this method, as we have only confirmed the technical conditions of pyro-genotyping using the plasmid with a limited number of clinical samples from volunteers. A prospective study using the mouthwash samples of HPV-positive and negative patients with head and neck cancers as well as analyses using our pyro-genotyping and conventional techniques is warranted.

## 4. Materials and Methods

### 4.1. Characteristics of the Patient Samples

In total, 52 patient volunteers who underwent otolaryngoscopic examination for health screening, cold, allergic rhinitis, and pollen allergy (exclusion criteria: none had head and neck cancer or were treated for oral HPV infection before) at the Toho University Omori Medical Center (Tokyo, Japan) from April 2018 to December 2018 were recruited for the study. Fifty milliliters of mouthwash waste fluids were collected from the waste tank with their consent.

The study protocol followed the ethical guidelines of the 2013 Declaration of Helsinki and was approved by and performed in accordance with all the relevant guidelines and regulations of the Toho University Institutional Review Board (approval code, A18069; approval date, 17 October 2018). Informed consent was obtained from each patient. Details of the study were uploaded in UMIN-CTR (#UMIN000037959) https://www.umin.ac.jp/ctr/index.htm.

### 4.2. Collection of Mouthwash

In the outpatient otolaryngology clinics, the otolaryngologist washed the patient’s mouth and suctioned the wash fluid using an otolaryngoscopic device. The mouthwash waste fluid of multiple patients was collected and pooled at the end; specimens were obtained each day. The collected mouthwash waste fluid samples were immediately centrifuged at 1800× *g* for 10 min, and the pellets were frozen at −80 °C. The DNA was extracted from the pellets using the standard phenol-chloroform method. The DNA concentration and quantity were measured using a NanoDrop spectrophotometer (ND-1000 Spectrophotometer (Nano Drop Technologies, Wilmington, DE, USA)).

### 4.3. Construction of Plasmid DNAs Containing the Variable Sequence of the 18 HPV Genotypes

The sequences of the 18 HPV genotypes (6, 11, 16, 18, 31, 33, 35, 39, 40, 42, 43, 44, 45, 52, 56, 58, 59, and 66) downstream of the HPV L1 MY09/11 consensus HPV genotyping primer site were artificially constructed, and the sequence lengths were between 449 and 455 bp. The sequences were aligned using the Omiga program, version 1.1.3 (Oxford Molecular Group, The Medawar Centre, Oxford, United Kingdom). The tree view of the pairwise alignment of our sequences was built based on their distance matrix in each genotype. The global alignment of each sequence was based on its free end gaps (65% similarity (5.0/−4.0), and it was performed using the genetic distance model of “Tamura-Nei” and the tree view model of “neighbor joining” using the Geneious prime 1.8.0_171-b11 sequence analysis software (Biomatters Ltd. Auckland, New Zealand) (Figure 1A,B). Each sequence was integrated into a plasmid (Blue Heron pUCKan, Eurofins Genomics, Tokyo, Japan) for direct sequencing and quantitative pyrosequencing.

### 4.4. Direct Sequencing Using Plasmid DNAs Harboring the 18 HPV Genotypes from HeLa Cells

Direct sequencing was performed after PCR amplification using specific primers. 1st PCR: 452 bp; sense-primer: MY09, CGTCCMARRGGAWACTGATC; antisense-primer: MY11, GCMCAGGGWCATAAYAATGG; 2nd PCR: 150 bp; sense-primer: GP5+, TTTGTTACTGTGGTAGATACTAC; antisense-primer: GP6+, GAAAAATAAACTGTAAATCATATTC. Touchdown PCR was used for the assays. All PCR methods included a denaturation step at 95 °C for 30 s, followed by an annealing step at various temperatures for 30 s, and an extension step at 72 °C for 30 s. The direct sequencing reaction was performed using a BigDye terminator kit (Applied Biosystems, Waltham, MA, USA), and the DNA of the 16 HPV genotypes was sequenced using an ABI 3100 automated sequencer (Applied Biosystems, Waltham, MA, USA) (Figure 2A–C).

### 4.5. Quantitative Pyrosequencing Analysis Using Plasmid DNAs Harboring the 18 HPV Genotypes from HeLa Cells and Clinical Mouthwash Samples

Before pyrosequencing analysis, we checked the concentration and quantity of the extracted DNAs of each sample using a NanoDrop spectrophotometer (ND-1000 spectrophotometer, Thermo Fisher Scientific KK, Tokyo, Japan). Appropriate primer sets were selected using a PyroMark assay design software version 2.0.1.15 (PyroMark assay Design, Qiagen, Valencia, CA, USA).

Pyrosequencing (PyroMark Q24, Qiagen) was performed after amplification and biotinylation using specific primers, which are as follows: 1st PCR 452 bp; sense-primer: HPV_MY09F_PYRO_biotin, CGTCCMARRGGAWACTGATC; antisense-primer: HPV_MY11R_PYRO, GCMCAGGGWCATAAYAATGG; HPV_Seq-Primer_PYRO, TGATTTACAYTTTATTTTTC. Touchdown PCR was used for the assays. All PCR methods included a denaturation step at 95 °C for 30 s, followed by an annealing step at various temperatures for 30 s, and an extension step at 72 °C for 30 s. The appropriate single bands after 1st and 2nd PCRs were confirmed using electrophoresis on 2.5% agarose gel. The pyrosequencing technique could quantitatively measure each nucleotide in the SEQ (unknown sequences) mode (sequence to analyze: MiX163140_AssaySetUpNo.12: tactctcgacatgatcgtgcactgtatacgtgctatactgcagtagtagtcagtactgatagtagcatgtactactgta) (Figure 1A,C). The quality of the pyrosequencing analysis was confirmed using the PyroMark Q24 Advanced software version 3.0.0 (PyroMark Q24 Advanced, Qiagen, Valencia, CA, USA) and the peak height threshold was adjusted to level 4 (required peak height for acceptance), A-peak reduction factor to 0.90, and quality control window to 20 (number of bases). Diluted water was used as the negative control, and genomic DNA extracted from Hela cells as the positive control.

### 4.6. HPV Detection using the Conventional HPV-DNA Method and Clinical Mouthwash Samples

A conventional HPV-DNA analysis (hybrid capture method) was performed using the Hybrid Capture 2 (HC2) modular system (DM3000, Qiagen) using the same clinical mouthwash samples as those used for pyrosequencing analysis (Table 1) [30].

### 4.7. HPV Detection using the Conventional HPV Genotyping Method and Clinical Mouthwash Samples

A conventional genotyping analysis using a restriction fragment length polymorphism (RFLP; RsaI, DdeI, HaeIII) assay (SRL Tokyo, Japan) was performed according to the manufacturer’s instructions using the same clinical mouthwash samples as those used for pyrosequencing analysis (Table 1).

### 4.8. Statistical Analysis

All statistical analyses were performed using the SPSS for Windows, version 12 (SPSS, Inc, Chicago, IL, USA) and the PRISM software for Windows, version 7 (GraphPad Prism, Inc, San Diego, CA, USA). Tree view analyses were performed using the Geneious prime 1.8.0_171-b11 sequence analysis software (Biomatters Ltd. Auckland, New Zealand). The categorical variables were compared using the Chi-square test and Fisher’s exact test, as appropriate. All reported *p* values were 2-sided, and *p* < 0.05 was considered statistically significant.

## Figures and Tables

**Figure 1 ijms-21-03697-f001:**
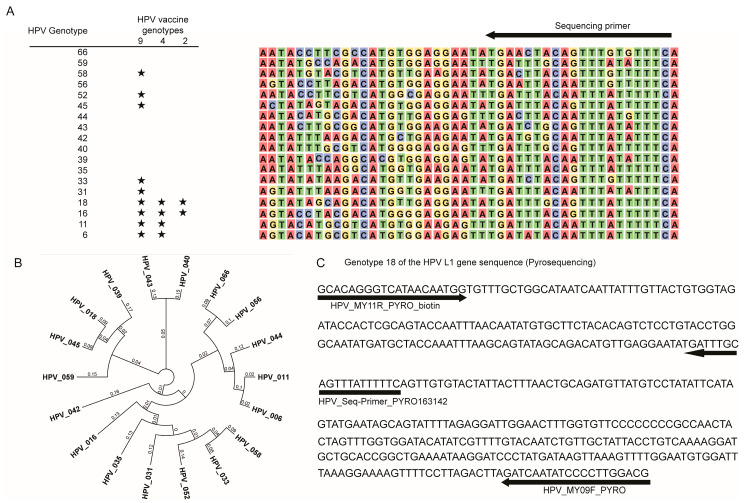
Eighteen genotypes based on the human papillomavirus (HPV) sequence in the L1 region, nucleotide differences, and primer pairs. (**A**) HPV L1 gene sequences in each of the 18 genotypes (arrows: primer pairs, stars: genotypes of each of the three HPV vaccines). (**B**) Tree view of HPV L1 pairwise alignment based on building distance matrix by free end gaps (65% similarity (5.0/−4.0)) in each of the 18 genotypes. Distance was obtained from pairwise alignments of all sequence pairs. The genetic distance model was generated using the Tamura-Nei algorithm of Geneious prime 1.8.0_171-b11. (**C**) Location of the sense/antisense and sequencing primers for genotype 18 of the HPV L1 gene. Black bold arrows are primer locations for pyrosequencing.

**Figure 2 ijms-21-03697-f002:**
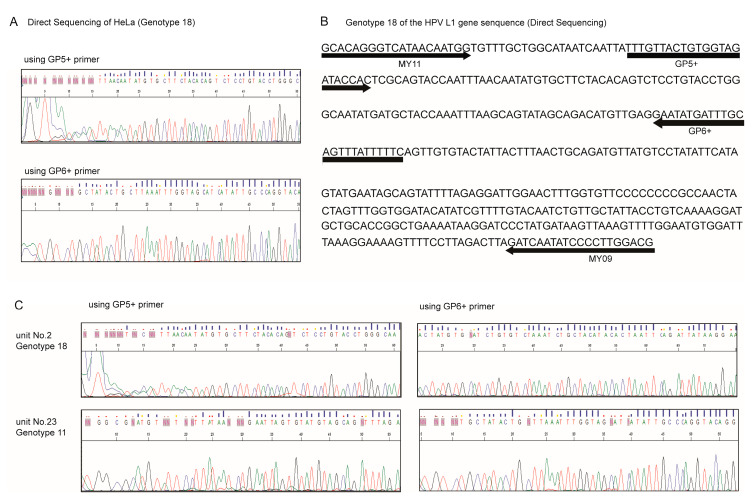
HPV L1 gene sequence in HeLa cells and clinical samples. (**A**) Direct sequencing analysis of HeLa cells using the GP5+/GP6+ primer pairs. (**B**) Primer information for direct sequencing. Black bold arrows are primer locations for direct sequencing. (**C**) Direct sequencing analysis of HPV-positive clinical samples (unit no. 2: genotype 18, unit no. 23: genotype 11).

**Figure 3 ijms-21-03697-f003:**
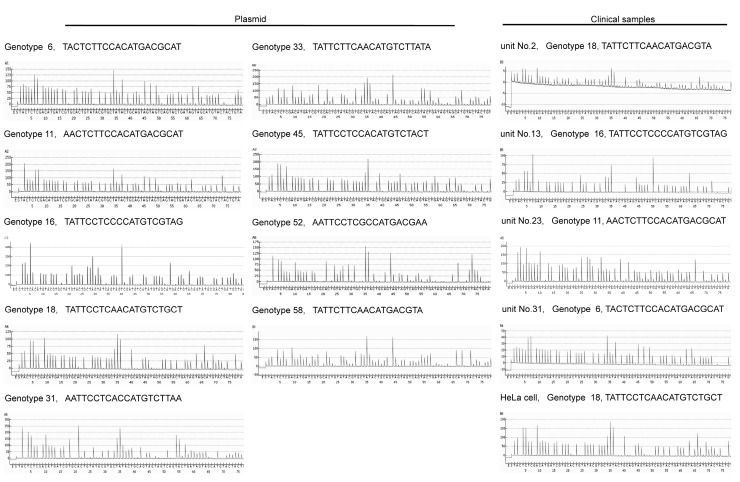
Pyrograms of nine plasmid HPVs (genotype 6, 11, 16, 18, 31, 33, 45, 52, and 58) and HPV-positive clinical samples.

**Table 1 ijms-21-03697-t001:** Summary of the results obtained using three methods on clinical samples.

Unit No.	DNA Conc. (ng)	Pyrosequencing		Restriction Enzyme-Based Genotyping
Amplification	HPV Genotype	Hybrid Capture
1	811.9				
2	146.1	Amplified	18		18
3	156.7				
4	708.7				
5	67.8				
6	117.2	Amplified	18		Unidentified
7	275.5				
8	153.1				
9	456.2				
10	612.4				
11	104.5				
12	586.7				
13	367.6	Amplified	16	Detected	16
14	128.8	Amplified	18		Unidentified
15	385.1				
16	327.5				
17	189.9				
18	252.4				
19	876.1				
20	8.4				
21	290.9				
22	37.4				
23	344.7	Amplified	11		Unidentified
24	26.1				
25	344.8				
26	3.8				
27	0.9				
28	197.3				
29	431.3				
30	301.1				
31	280.6	Amplified	6		Unidentified
32	275.2				
33	486.1				
34	380.2	Amplified	18		Unidentified
35	298.4	Amplified	18		Unidentified
36	260.3				
37	280.8				
38	180.2				
39	302.1				
40	197.6				
41	245.2				
42	266.1				
43	407.8				
44	28.1				
45	180.3				
46	199.1				
47	240.6	Amplified	18		Unidentified
48	150.2	Amplified	18		Unidentified
49	118.2				
50	260.8	Amplified	18		Unidentified
51	110.8				
52	320.1

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
