# Peer review of "Mouthwash-Based Highly Sensitive Pyro-Genotyping for Nine Sexually Transmitted Human Papilloma Virus Genotypes"

_ijms, 2020, doi:10.3390/ijms21103697_

Round 1
Reviewer 1 Report
Dear Authors,
in my opinion the manuscript is worthy for publication on IJMS.
Best regards
Author Response
Dear Reviewer 1:
I, along with my coauthors, wish to resubmit the attached manuscript for publication in International Journal of Molecular Science, titled “Mouthwash-Based Highly Sensitive Pyro-Genotyping for Nine Sexually Transmitted Human Papilloma Virus Genotypes” The paper was coauthored by Yukiko Seto, Ritsuko Oikawa, Takara Nakazawa, Hanae Furuya, Hidehito Matsui, Sachiko Hosono, Mika Noike, Akiko Inoue, Hiroyuki Yamamoto, Fumio Itoh, and Kota Wada.
Manuscript ID: ijms-817063
Type of manuscript: Article
>in my opinion the manuscript is worthy for publication on IJMS.
We really appreciate it.
Sincerely,
Yoshiyuki Watanabe
Department of Internal Medicine, Kawasaki Rinko General Hospital
Kawasaki, Kanagawa 210-0806, Japan
+81-44-977-8111
+81-44-976-5805
ponponta@marianna-u.ac.jp

Reviewer 2 Report
The authors successfully addressed the raised issues. However, the lower psychological resistance of the proposed test as compared to gynecological visit is confusing and should be deleted since the two methods analyze HPV presence in completely different cancers and in diverse anatomic location.
Author Response
Dear Reviewer 2:
I, along with my coauthors, wish to resubmit the attached manuscript for publication in International Journal of Molecular Science, titled “Mouthwash-Based Highly Sensitive Pyro-Genotyping for Nine Sexually Transmitted Human Papilloma Virus Genotypes” The paper was coauthored by Yukiko Seto, Ritsuko Oikawa, Takara Nakazawa, Hanae Furuya, Hidehito Matsui, Sachiko Hosono, Mika Noike, Akiko Inoue, Hiroyuki Yamamoto, Fumio Itoh, and Kota Wada.
Manuscript ID: ijms-817063
Type of manuscript: Article
The authors successfully addressed the raised issues. However, the lower psychological resistance of the proposed test as compared to gynecological visit is confusing and should be deleted since the two methods analyze HPV presence in completely different cancers and in diverse anatomic location.
We really appreciate the advices. We agree with you and we deleted it.
Again, thank you very much.
Sincerely,
Yoshiyuki Watanabe
Department of Internal Medicine, Kawasaki Rinko General Hospital
Kawasaki, Kanagawa 210-0806, Japan
+81-44-977-8111
+81-44-976-5805
ponponta@marianna-u.ac.jp

This manuscript is a resubmission of an earlier submission. The following is a list of the peer review reports and author responses from that submission.
Round 1
Reviewer 1 Report
Dear Authors,
your manuscript is a well written article that analyze the efficacy of the Mouthwash- based highly Sensitive HPV Pyro-2 Genotyping in the diagnosis of HPV infections.
I believe that, despite the limited sample enrolled, the research demonstate the efficacy, in comparison with other methods, of this less expensive, high-throughput, simple method for diagnosins STI.
For these reasons, I believe that manuscript could be accepted for publciation on IJMS.
Best Regards
Author Response
April 03, 2020
Dear Reviewer 1:
I, along with my coauthors, wish to resubmit the attached manuscript for publication in International Journal of Molecular Science, titled “Mouthwash-based highly Sensitive 9 HPV Pyro-Genotyping for Sexually Transmitted HPV Infections.” The paper was coauthored by Yukiko Seto, Ritsuko Oikawa, Takara Nakazawa, Hanae Furuya, Hidehito Matsui, Sachiko Hosono, Mika Noike, Akiko Inoue, Hiroyuki Yamamoto, Fumio Itoh, and Kota Wada. The manuscript ID is ijms-747829.
We appreciate the critical advices from you and agree with your constructive comments. We understood that our sample size was limited but also we stated excessive opinion in our manuscript without any verifications such as the positivity of mouth waste fluid associated with the positivity of the cervical mucous, or vice versa.
We have made the following revisions to our manuscript:
- We focused on oral HPV detection using our new methodology (Pyro-Genotyping).
- We asked an English language editing company to revise our manuscript for the language.
We thank you for your consideration and look forward to hearing from you.
Sincerely,
Yoshiyuki Watanabe
Department of Internal Medicine, Kawasaki Rinko General Hospital
Kawasaki, Kanagawa 210-0806, Japan
+81-44-977-8111
+81-44-976-5805
ponponta@marianna-u.ac.jp

Reviewer 2 Report
This manuscript describes the implementation of a mouthwash-based test coupled with pyrosequencing for the detection of nine HPVs in oral rinse specimens of healthy volunteers for screening purposes. Despite the idea is potentially interesting, since could lead to the identification of subject at risk of developing HPV cancers with a simple test, the manuscript as it stands is difficult to read due to a poor English and scientific terminology. For example, the extensive use of quotation marks and colloquial sentences does not sound as scientifically appropriate. A substantial revision of the outline of the article is required.
In addition, the idea is not really novel, since several of papers already report such kind of approach (for example Rosenthal et al, 2017; Fakhry et al, 2019) successfully identifying more than 9 HPV types by PCR-based methods in oral rinse of HNC patients. Therefore, the novelty of the present paper is low.
Also, several issues must be addressed by the authors:
- It is unclear why the authors propose their method as cervical cancer screening test, since in cervical cancer HPV infection should be confined in cervix and not in oral regions
- The methods for collecting specimens in not clearly described: in particular, is each sample collected and stored immediately and then pooled? Or the collection and pooling was done at the end, when all specimens were obtained?
- The authors must clarify why they pooled samples instead of performing a more reasonable single-subject analysis.
- To test sensitivity of their approach, the authors compare their methods with Hybrid Capture and Restrict enzyme-based genotyping. To proof that their pyrosequencing is more accurate than the currently used approach to detect HPV DNA, the authors should perform a comparative analysis with PCR-based approaches such as the ones described in the previous mentioned papers.
Author Response
April 03, 2020
Dear Reviewer 2:
I, along with my coauthors, wish to resubmit the attached manuscript for publication in International Journal of Molecular Science, titled “Mouthwash-based highly Sensitive 9 HPV Pyro-Genotyping for Sexually Transmitted HPV Infections.” The paper was coauthored by Yukiko Seto, Ritsuko Oikawa, Takara Nakazawa, Hanae Furuya, Hidehito Matsui, Sachiko Hosono, Mika Noike, Akiko Inoue, Hiroyuki Yamamoto, Fumio Itoh, and Kota Wada. The manuscript ID is ijms-747829.
We appreciate the critical advices from you and agree with your constructive comments. We understood that we stated excessive opinion in our manuscript without any verifications such as the positivity of mouth waste fluid associated with the positivity of the cervical mucous, or vice versa.
We have made the following revisions to our manuscript:
- This manuscript describes the implementation of a mouthwash-based test coupled with pyrosequencing for the detection of nine HPVs in oral rinse specimens of healthy volunteers for screening purposes. Despite the idea is potentially interesting, since could lead to the identification of subject at risk of developing HPV cancers with a simple test, the manuscript as it stands is difficult to read due to a poor English and scientific terminology. For example, the extensive use of quotation marks and colloquial sentences does not sound as scientifically appropriate. A substantial revision of the outline of the article is required.
We appreciate the critical advices. We agree with your constructive comments. We understood that we stated excessive opinion in our manuscript without any verifications such as the positivity of mouth waste fluid associated with the positivity of the cervical mucous, or vice versa. We changed outline as “oral HPV detection using our new methodology (Pyro-Genotyping)”. We deleted comments about any “mouth waste fluid” and “cervical mucosa”, because we did not have any evidence for these. We also change all “Abstract, Introduction and Discussion” of our manuscript. We also asked an English language editing company to revise our manuscript for the language.
- In addition, the idea is not really novel, since several of papers already report such kind of approach (for example Rosenthal et al, 2017; Fakhry et al, 2019) successfully identifying more than 9 HPV types by PCR-based methods in oral rinse of HNC patients. Therefore, the novelty of the present paper is low.
We appreciate the advices We agree with you. Rosenthal already reported 14 HPV genotyping using oral rinse by realtime PCR-based Cobas® HPV Test (Roche Diagnostics), but not includes 6 and 11 genotypes. Moreover, Cobas® HPV Test needs 12 different fluorescent DNA probes and 16 different primers and determine genotype 16 or 18 or others (not specific). Fakhry also already reported HPV genotyping using oral rinse or swab samples by in situ hybridization or PCR technique and determine genotype16 or other oncogenic HPV (not specific). Our Pyro-Genotyping is available not only high risk HPV genotype (specific) but also 6 and 11, using unique primer and probe. However, we understood that we should comment these reports in our manuscript. Thank you so much your advice.
- It is unclear why the authors propose their method as cervical cancer screening test, since in cervical cancer HPV infection should be confined in cervix and not in oral regions.
We appreciate the critical advices We totally agree with you. We deleted comments about any association between “mouth waste fluid” and “cervical mucosa”, because we did not have any evidence for these.
- The methods for collecting specimens in not clearly described: in particular, is each sample collected and stored immediately and then pooled? Or the collection and pooling was done at the end, when all specimens were obtained?
I agree with you. We describe the detail information of sample collection in our manuscript. During outpatient otolaryngology clinics, otolaryngologist washed patient’s mouth and suctioned using otolaryngoscopic device. Collection and pooling of the mouthwash waste fluids of multiple patients were done at the end, all specimens were obtained each day. The collection of mouthwash waste fluids samples was immediately centrifuged and the pellets were frozen at -80°C.
- The authors must clarify why they pooled samples instead of performing a more reasonable single-subject analysis.
We agree with you. We describe the reason in our manuscript. We made plan to collect mouthwash waste fluids from each patient. However, it is difficult to connect collecting tube between otolaryngoscopic device system technically. We finally decide to analyse HPV genotyping using pooled samples at the first step of our research. After that, we will ask otolaryngoscopic device company and make plan of single-subject analysis near the future.
- To test sensitivity of their approach, the authors compare their methods with Hybrid Capture and Restrict enzyme-based genotyping. To proof that their pyrosequencing is more accurate than the currently used approach to detect HPV DNA, the authors should perform a comparative analysis with PCR-based approaches such as the ones described in the previous mentioned papers.
We agree with you. In Japan, we usually use Hybrid Capture and Restrict enzyme-based genotyping with insurance coverage. Now a days, we also can use Cobas® HPV Test (PCR based method), but expensive than Pyro-Genotype because of Cobas® HPV Test needs multi-fluorescent DNA probes. Moreover, it can determine genotype 16 or 18 or others (not specific). But we agree with you. We will also try to compare PCR-based analysis using single-subject analysis near the future.
We thank you for your consideration and look forward to hearing from you.
Sincerely,
Yoshiyuki Watanabe
Department of Internal Medicine, Kawasaki Rinko General Hospital
Kawasaki, Kanagawa 210-0806, Japan
+81-44-977-8111
+81-44-976-5805
ponponta@marianna-u.ac.jp

Round 2
Reviewer 2 Report
The authors successfully addressed most of the raised issues, even if the article is still difficult to read and not really well written. Working on the clarity and the outline of the manuscript is again strongly suggested.
In addition, there is still an inappropriate link of their proposed method with cervical cancer (Line 161: “feel resistance towards gynecological examination”) that should be deleted.
Also, a couple of further issues should be addressed. How do they check the quality/integrity of their genomic preparation before and during the pyrosequencing? Importantly, a better description of the volunteer’s cohort should be provided. Are they all healthy patients? Do they suffer of any HPV-related disease? The validation of the proposed method should be compared to currently available techniques in at least in few patients with a diagnosis of HPV-positive and HPV-negative HNC.